# Feeding Sows with Multi-Species Probiotics During Late Pregnancy and the Lactating Period Influences IgA Concentration in Colostrum and Subsequently Increases the Survival Rate of Piglets in Porcine Epidemic Diarrhea Outbreak Herd

**DOI:** 10.3390/ani15010103

**Published:** 2025-01-05

**Authors:** Narathon Innamma, Kampon Kaeoket

**Affiliations:** Department of Clinical Sciences and Public Health, Faculty of Veterinary Science, Mahidol University, Phuttamonthon, Nakhon Pathom 73170, Thailand; narathon.inn@student.mahidol.edu

**Keywords:** sow colostrum, immunoglobulin A, porcine epidemic diarrhea, pre-weaning mortality rate, probiotics

## Abstract

Increasing the immunoglobulin A (IgA) potential of sow colostrum protects newborn piglets against infection during the pre- and post-weaning periods. Feeding pigs with multi-species probiotics (5 g/sow/day) via top dressing from 4 weeks before farrowing until weaning increases the IgA levels in colostrum during the first 6 h after farrowing. This subsequently improved the piglets’ weaning weight and reduced the pre-weaning mortality rate in an outbreak breeder herd with porcine epidemic diarrhea (PED).

## 1. Introduction

Porcine epidemic diarrhea (PED) virus is an important cause of diarrhea in sows and piglets. Particularly in piglets, this porcine coronavirus causes severe damage to the small intestine by decreasing the proportion of crypt and villi from 1:7 to 1:3, subsequently driving the malabsorption of nutrients and electrolytes [1]. This is the primary cause of the high mortality rate (i.e., 20–50%) and growth retardation in piglets in infected herds, leading to economic losses in the pig industry worldwide. In the United States, the PEDV outbreak in 2013–2014 caused the pig population to decrease by approximately 3.2% compared to the previous year [2,3,4]. Nowadays, the most practical method is to provide passive immunity to piglets via colostrum from immunized sows. There are two well-documented techniques to immunize sows: vaccination of gilts at replacement unit and before farrowing and feeding the sows fresh small intestines from sick piglets with PED [4,5]. Using these techniques, sow colostrum will develop passive immunity or maternally derived antibodies (MDAs), especially Immunoglobulin G (IgG) and Immunoglobulin A (IgA) specific to PEDV [6,7]. In practice, in order to promote piglet health, every single piglet must intake colostrum from its mother as soon as possible after birth, since it is enriched with IgG and IgA. This is absorbed via the small intestine only for the first 24 h of life, the so-called “gut closure” period [6]; thus, to obtain the highest quantity, every single piglet should intake colostrum during the first 2–6 h of life. This is because IgG and IgA production in sow colostrum reaches the maximal level at about 2 h and this is maintained for 6 h; thereafter, the amounts of IgG and IgA decrease dramatically [8,9]. In normal sows, the average value of IgA in colostrum at birth is 23.8 mg/mL, and this decreases to 7.85 mg/mL at 6 h and to 4.59 mg/mL at 24 h after the onset of farrowing [10]. In addition, IgA localized at the surface of small intestines of piglets causes a mucosal barrier/mucosal immune response in order to prevent pathogen attachment [11]. Therefore, in practice, during an outbreak of PED, if a pig farmer can find a strategy to promote high IgA production in colostrum, this certainly guarantees that all the piglets keep their guts healthy.

Probiotic bacteria, the friendly bacteria of the gut, have multiple and various influences on the host, e.g., different organisms can influence the intestinal luminal environment, epithelial and mucosal barrier functions, and the mucosal immune system [12]. In a previous study, a multi-strain probiotic (10^11^ CFU/g of *Lactobacillus acidophilus*, *Bifidobacterium bifidum*, *Lactobacillus delbrueckii*, *Bacillus subtilis*, and *Lactobacillus rhamnosus*) had an overall positive effect on feed intake, the feed conversion ratio, and the IgG and IgG antibodies of Japanese quails and decreased the amount of *Esherichia coli* (*E. coli*) [13]. In pig production, probiotics increase the milk yield of lactating sows, enhance the welfare of pregnant sows, improve the growth of nursery pigs, and control diarrhea-causing pathogens in pig farms [14,15]. BACTOSAC-P™ (KMP Biotech Co. Ltd., Thailand) is a commercial multi-species probiotic product that contains 1.0 × 10^7^ CFU/g of seven probiotic bacteria; these are *Lactobacillus acidophilus*, *Lactobacillus plantarum*, *Streptococcus faecium*, *Pediococcus pentosaceus*, *Bacillus subtilis*, *Bacillus licheniformis*, and *Saccharomyces cerevisiae*. The components of BACTOSAC-P™ are harvested from nature in Southeast Asia, and each dose has an equal concentration of the probiotics. BACTOSAC-P™ contains the selected probiotic bacteria, which can produce lipase enzymes to digest lipids in the intestines of hosts [16]. With these properties in mind, we hypothesize that BACTOSAC-P™ may facilitate an increase in IgA yield in colostrum in a sow herd during PED infection.

The aim of the present study was to investigate the potential of BACTOSAC-P™ to promote Immunoglobulin A-containing-colostrum production in sows during 24 h of lactation and subsequently promote piglet growth and diminish the pre-weaning mortality rate in a porcine epidemic diarrhea virus (PEDV)-infected herd.

## 2. Materials and Methods

### 2.1. Ethics Statement

This research project was approved by the Faculty of Veterinary Science—Institute Animal Care and Use Committee (FVS-IACUC-Protocol No. MUVS-2018-06-26), Mahidol University, Thailand.

### 2.2. Animals

This experiment was performed on 20 sows (primiparous and multiparous) in a pig farm located in Saraburi, Thailand. The animals were reared in a continuous farrowing system. The pregnant sows were housed in individual gestation crates. During lactation, the pigs remained in the individual stalls. This pig breeding farm was confirmed to be contaminated by PEDV in the middle of May 2019 by using antigen screening test kits (Bionote, Gyeonggi-do, Republic of Korea) and the PCR technique, and the experiment was carried out in the middle of June 2019. Altogether, 20 sows from the PED-infected herd, with an average parity number of 2.4 ± 1.4, were included in this study. At 12 weeks of pregnancy, all the sows were fed the minced small intestines from PED-infected piglets following the farm’s protocol. The sows’ gestation and lactation basal diets were formulated to meet the nutrient requirements of swine recommended by National Research Council [17]. The formulation and nutrient specifications of the gestating and lactating sows’ diets are shown in Table 1. The sows were fed during the first, second, third, and fourth months of pregnancy approximately 2.0–2.5, 3.0–3.5, and 3.5 kg feed per sow per day, respectively. However, a week before parturition, they were fed 3 kg feed per sow per day. After parturition, the sows were fed 3.0 kg of feed per day, and the amount of feed offered to the sows was increased by 0.5 kg per day until they were fed ad libitum (6.0–7.0 kg) from week one of lactation until weaning according to the farm’s protocol. Thereafter, they were randomly divided into 2 groups (i.e., control and treatment), with 10 in each group. They were kept in the same farrowing house equipped with an evaporative cooling system and given normal feed (control group) or a mixture of BACTOSAC-P™ (5 g/sow/day), which is the concentration recommended by the supplier (K.M.P.BIOTECH CO., LTD., Thailand), via top dressing on normal feed (treatment) from 4 weeks before farrowing until weaning (i.e., lactation period of 24 days). On day 113 of pregnancy, in order to take good care of the piglets, all the sows were induced via the injection of 5 mg Dinoprost into the perivulva area either at the 3 or 9 o’clock position as previously described [18,19]. All the sows were then farrowed at 24 h after induction between 7 and 10 am.

### 2.3. Colostrum and Milk Sample Collection

The time when the first piglet was born was designated as 0 h. Colostrum and milk samples were collected from each sow in each group, considering the convenience of sample collection and the safety of both the animals and the personnel. A portion of colostrum were collected from all the teats as a pool sample (approximately 50 mL) at 3, 6, 12, and 24, and a portion of milk was collected from all the teats as a pool sample (approximately 50 mL) at 48 h, and then kept in sealed container and stored at −20 °C until analysis [9,20].

### 2.4. Measurement of Pig Immunoglobulin A (IgA) Level

The Immunoglobulin A (IgA) level was measured by using a pig IgA ELISA kit (Koma Biotech Inc., Seoul, Republic of Korea). Analysis was performed according to the manufacturer’s instructions. Briefly, all the samples were diluted 100 times before measurement. The ELISA procedure involved the following steps: (1) coating: 100 μL of the diluted coating antibody was added to each well, which was then covered and incubated at 4 °C overnight. (2) Washing: the wells were washed four times with 300 μL of washing solution and the excess liquid was removed after the last wash. (3) Blocking: 200 μL of blocking solution was added per well and incubated at room temperature for 1 h; then it was washed again, as in step 2. (4) Reaction: 100 μL of standard, blank, or sample was added to each well in duplicate and incubated for 1 h at room temperature. (5) Detection: 100 μL of diluted detection antibody was added to each well and incubated for 1 h at room temperature, followed by washing. (6) Color Development: 100 μL of TMB or pink-ONE TMB solution was added to each well and the color was allowed to develop. (7) Stop Reaction: 100 μL of stop solution was added to each well and the absorbance was measured at 450 nm using a microplate reader. The results were recorded in ng/mL, and then the data were calculated in mg/mL [9].

### 2.5. Measurement of Backfat Thickness of Sows

The sows’ backfat thickness was measured by using a digital backfat indicator (Renco Corp., Minneapolis, MN, USA). The average value from both sides of the P2 position (6.5 cm away from body midline at the last rib level) was used as the backfat thickness (mm) [21]. The sows’ backfat thickness was measured at 4 weeks before farrowing and at weaning (day 24 of lactation).

### 2.6. Production Performance Parameters

Sow reproductive performance measurements included total number of piglets born, born alive, stillborn, mummified, and weaned per litter. Within 12 h of birth, the litter birth weights of these piglets were weighed and recorded. All piglets were processed according to the standard operating procedure established by the farm within 24 to 48 h of birth. Piglet processing included tail docking, needle teeth clipping, administering injectable iron, and castration of male piglets. Incidence of stillborn and mummified piglets was recorded at birth. Any pigs that died shortly before or during parturition, due to asphyxia or dystocia, were classified as stillborn. Piglets were monitored daily for instances of morbidity and mortality. Any dead piglets were recorded by date of death. Pre-weaning mortality in piglets per litter was calculated as a percentage, based on the number of piglets that die between birth and weaning, by the following formula:Preweaning Mortality per litter (%) = (Number of piglet deaths before weaning in each litter/Total number of piglets born alive in each litter) × 100

One day before weaning, individual piglet body weights were determined and recorded to calculate total weight gain during the pre-weaning period. Piglets were weaned at about 24 ± 1.0 day of age.

### 2.7. Statistical Analysis

All data were tested for normality prior to analysis by examination of histograms and normal distribution plots using the Shapiro–Wilk Test. The IgA levels in the groups were analyzed at different timepoints by using IBM SPSS Statistics for Windows, version 26.0 (SPSS Inc., Chicago, IL, USA). ANOVA was used to analyze the pig IgA levels across the different timepoints in the groups, and we compared the means by using Duncan’s multiple range test for a stepwise comparison approach, which can be more effective for identifying subtle differences in means. A T-test was used to analyze the production performance parameters between the groups. A statistically significant difference was defined as *p* ≤ 0.05.

## 3. Results

According to the clinical findings, none of the sows in either of the groups showed clinical signs of PED when they were kept in the farrowing house. However, their piglets started showing clinical signs of PED at 4 days old. The pig IgA levels in colostrum in the control and treatment groups are shown in Figure 1. In the treatment group, the highest level of IgA was found in the sows fed BACTOSAC-P™ at 6 h (26.22 ± 7.09 mg/mL), the lowest level was found at 24 h (11.87 ± 11.58 mg/mL) (*p* < 0.001), and the IgA level of 4.51 ± 2.84 mg/mL in sow milk was found. In the control group, the highest level of IgA was found at 3 h (16.16 ± 2.50 mg/mL), and the lowest level was found at 24 h (3.41 ± 2.44 mg/mL) (*p* < 0.001) and the IgA level of 3.41 ± 2.44 mg/mL was found in sow milk. According to the comparison between the total IgA levels in colostrum and milk across the different timepoints within and between the groups in Figure 2, the pig IgA levels in both the groups from 3 to 48 h were higher in the treatment group than in the control group, especially at 6 h (*p* = 0.10).

The backfat thicknesses of both sow groups are shown in Table 2. There was no significant difference in terms of the backfat thickness of both the sow groups at the start of the experiment (1 month before farrowing). However, there was a significant difference in the backfat thickness of the sows 3 weeks after farrowing. The treatment-administered sows showed significantly thicker backfat (11.70 ± 0.14 mm) than those in the control group (11.13 ± 0.17 mm) (*p* < 0.05).

The production performance parameters are shown in Table 3. There was no significant difference in terms of the number of total piglets born per litter, the number of piglets born alive per litter, and the litter birth weight. However, the pre-weaning mortality rate was two times higher in the control group (53.6%) than that in the treatment group (24.9%). The same case was also found for the number of piglets weaned per litter. In addition, a significantly higher weaning weight (5.9 kg) was found in the treatment group than that in the control (3.9 kg) (*p* < 0.05).

## 4. Discussion

PEDV remains a significant health issue, causing high mortality in pre-weaning piglets and economic losses [1,2,3]. Co-infection porcine deltacoronavirus with PDCoV exacerbates disease severity by increasing viral shedding, disrupting intestinal structure, and elevating infection levels. Limited vaccine efficacy, likely due to immunization challenges and misdiagnosis, underscores the need for new antiviral strategies against PEDV [22,23]. Therefore, it is necessary to explore new antiviral strategies to reduce the infectivity of the pandemic strain of PEDV among pigs.

Recent literature data highlight the beneficial effects of administering probiotics to pigs, such as the regulation of the intestinal microflora, the inhibition of pathogens in the gastrointestinal tract, improved intestinal barrier function, and the enhancement of mucosal immunity. The supplementation of probiotic Lactobacillus fermentum I5007 in newborn piglets can regulate the formation of gut microflora and reduce the number of enteropathogenic *Escherichia* spp. and *Clostridium* spp. in the gastrointestinal tract [24]. Lactic acid bacteria have shown antiviral effects. The cell-free supernatant (CFS) refers to the liquid obtained after separating cells from a culture medium through processes like centrifugation or filtration. CFS consists of the substances secreted by the cells during cultivation, including proteins, enzymes, metabolites, and bioactive compounds [25]. The cell-free supernatant (CFS) of the *Lactobacillus* spp. probiotic and live *Lactobacillus plantarum* and *Pediococcus* spp. showed protective effects against the pandemic strain of PEDV [26,27]. The CSF of lactobacilli could reduce the viral infectivity of Vero cells, and the live Lactobacillus plantarum strain 25F reduced the cytopathic effect of PEDV. One possible mechanism may be the blocking of viral adsorption into the host cells by CFS metabolites, for example, organic acids, organic compounds (diacetyl), short-chain fatty acids, and antimicrobial peptides [28,29]. Lactic acid bacteria enable the transformation of complex nutrients, such as plant cell wall components (pectin, cellulose, and hemicellulose), into simple sugars that ferment into short-chain fatty acids (SCFAs), mainly acetate, propionate, and butyrate [30]. Microbial-derived short-chain fatty acids are crucial for protecting the intestinal barrier and regulating the immune system to respond to viral infection [31]. Several experiments demonstrated that adding short- and medium-chain fatty acids (MCFAs) can effectively combat viral infection [32,33,34,35,36,37]. Selected MCFAs, mainly caprylic, capric, and lauric acids, and a related monoglyceride, glycerol monolaurate (GML), can inhibit African swine fever virus when administered as liquid or feed [38]. In the case of PEDV, butyrate provides protection from PEDV infection in the intestinal epithelial cells. One possible mechanism may be the activation of the innate immune response by GPR43. A previous study suggested a strategy involving the inhibitory effect of G protein-coupled receptors against PEDV infection [39].

Lactic acid bacteria (LAB) probiotics have been shown to enhance the intestinal barrier. The administration of *Lactobacillus delbrueckii* (LAB) to piglets during the suckling period can improve their intestinal morphology and barrier function [40]. A study found that LAB administration can cause greater jejunal and ileal villus heights and a better villus–crypt ratio when compared to those of untreated piglets. In addition, the administration of LAB slightly increases mRNA expression for occlusion and the quantity of ZO-1 in the jejuna and ilia of piglets during the suckling period, which indicates that LAB can improve intestinal barrier integrity [41]. These data suggest that LAB promotes the gut health of piglets during weaning, and this might increase their body weight and improve their health status. Probiotic bacteria stimulate the immune systems of pigs by triggering gut-associated lymphoid tissue (or GALT)-related activities. This occurs by increasing the quantities of T lymphocyte cells in the intestinal mucosa and Immunoglobulin A (IgA), which represents one type of antibody production, bringing out disease resistance in hosts [42].

Increasing the immune potential of sow colostrum protects newborn piglets against infection during the pre- and post-weaning periods. These results clearly show that probiotic-containing bacteria, BACTOSAC-P™, have the potential to promote IgA-containing colostrum and milk production in sows for 48 h during lactation and also have a beneficial effect on piglet growth and reduce the pre-weaning mortality rate during PED outbreaks in breeding herds. Considering the level of IgA in the pig colostrum and milk, the sows fed BACTOSAC-P™ (5 g/sow/day) produced more IgA during the first 48 h of lactation, particularly at 2–6 h. Similar results have been obtained in previous studies [43,44,45]. The reason for there being a higher IgA level in the pig colostrum in the treatment group might be that probiotic bacteria, friendly bacteria in the gut, influence the intestinal luminal environment, epithelial and mucosal barrier function, and the mucosal immune system [12]. In practice, farmers should ensure that all piglets intake colostrum from their mother before the gut closure phenomenon, which usually occurs at 24–36 h [46,47]. Nevertheless, the IgA level in colostrum reached its peak at about 6 h after farrowing, and thereafter gradually declined. In our experiment, the level of IgA during the first 6 h after farrowing is two times higher compared with that of the control; consequently, the piglets in the treatment group consumed more IgA-containing colostrum than the control group did. IgA may localize at the surface of the small intestines of piglets, performing a mucosal barrier/mucosal immune response in order to prevent pathogen attachment [11]. This mechanism may, at least in part, explain the higher weaning weight and lower pre-weaning mortality rate of the piglets from the sows fed BACTOSAC-P™. The factors that are associated with piglet colostrum consumption include the number of piglets weaned per litter and the initial and gained weaning weights of the piglets. Once the piglets started to suckle, maternal passive immunity was transferred from the sows to their piglets via colostrum [48]. In this work, probiotic supplementation in sows significantly enhanced the concentration of IgA via colostrum to the piglets and improved the survival and growth performance of the piglets. Similar results have been obtained in previous studies [49].

The gut microbiota is a community of microorganisms that include bacteria, archaea, fungi, and viruses that live in intestinal environments. The gut microbiota impacts the performance and health status of the host. Previous research shows that the initial colonization of the gut microbiota in piglets occurs at least within the immediate prenatal period [50]. Some studies indicate that the sows’ microbiota and the rearing environment influence gut microbiota functionality and the composition of their offsprings. Exposure of piglets to sows’ vaginas and their pen environment, exposure of sows and piglets to antibiotics, dietary treatments, and the length of time between sow microbiota modulation cause some differences between the sows’ microbiota and that of their progeny. They also influence their immune system development, growth, and survival [51,52,53]. The importance of colostrum and milk in relation to gut microbiota development and piglet health has been documented [54]. Receiving colostrum and milk in the early lives of piglets is important for intestinal microbiota colonization and immune system development. In our study, we found that the sows supplemented with probiotics were healthy and could produce sufficient quantities of high-quality colostrum and milk for newborn piglets. In addition, the fact that the sows were constantly supplemented with probiotics may cause the transfer and shedding of beneficial microorganisms into the environment and their piglet’s intestines [55,56]. These factors can support initial intestinal microbiota colonization in order to improve piglets’ health and survival.

The probiotic-supplemented sows had thicker backfat compared to that of the control sows. Backfat thickness is a significant parameter for female pigs, which is associated with reproductive performance, for example, puberty attainment, the total piglets born (TB), the farrowing rate, and the period from weaning to the first interval. Moreover, backfat is a significant source of hormones related to puberty attainment, such as leptin, insulin-like growth factor-I (IGF-I), and progesterone (P4) [21].

In a previous study [57], PED infection caused the deterioration in the growth performance of piglets at the suckling period. In this study, the piglets of the sows that were fed via intestinal feedback to defend against PED and supplemented with probiotics had a greater average weight at weaning and a lower mortality rate than did those that were farrowed from the sows subjected to intestinal feedback, but no probiotic supplementation. This agrees with an earlier report stating that probiotics increase the milk yield of lactating sows, improve the growth rate in nursery pigs, and control diarrhea-causing pathogens in pig farms [14]. The probiotic blend used in this study, which includes species that produce lipase enzymes [16], was selected based on its potential to enhance gut health through multiple mechanisms. Lipase-producing probiotics have been shown to improve nutrient digestion and absorption, which are particularly relevant in sows and piglets that are susceptible to post-weaning diarrhea. Enhanced lipid digestion may alleviate the burden on the digestive system, thus reducing intestinal stress and improving overall gut health.

This study primarily focused on evaluating the key immune and reproductive parameters in sows and piglets, specifically IgA levels and reproductive performance. However, we acknowledge that the scope of the measured indicators was limited. While these outcomes provide important insights into the effects of multi-species probiotics, a more comprehensive assessment, including additional biomarkers, such as intestinal morphology, digestive enzyme activity, and microbial composition, could further strengthen our understanding of how probiotics influence the prevention of PED. These additional measures would allow us to evaluate the broader physiological impact of probiotics, particularly in relation to gut health, which is closely linked to the severity of PED. We recognize that this is a limitation of this study and recommend that future research incorporates a wider range of indicators to build upon these initial findings. While this study provides valuable insights into the effects of BACTOSAC-P™ on the reproductive performance and immunological response of lactating sows, it is important to acknowledge that this research primarily serves as initial verification of the probiotic product. The scope of this study was intentionally focused on evaluating the fundamental outcomes, such as the IgA levels and the piglet performance; however, we recognize that a more comprehensive exploration of the underlying mechanisms and broader physiological effects would strengthen these conclusions.

Future studies should aim to investigate additional biomarkers of gut health, including microbial composition, intestinal morphology, and cytokine profiles, to provide a more holistic understanding of how BACTOSAC-P™ interacts with the host. Moreover, exploring the dose–response relationships and the long-term effects of probiotic supplementation across the different stages of sow reproduction could offer deeper insights into the product’s efficacy and mode of action. By expanding the scope of investigation, subsequent research will be better positioned to elucidate the synergistic or antagonistic interactions among multi-species probiotics and uncover their precise roles in improving sow and piglet health.

## 5. Conclusions

Feeding pigs multi-species probiotics (5 g/sow/day) via top dressing from 4 weeks before farrowing until weaning increases immunoglobulin A (IgA) levels in colostrum during the first 6 h post-farrowing. This enhanced immune response contributes to improved weaning weight and a lower pre-weaning mortality rate in breeder herds during porcine epidemic diarrhea outbreaks. These findings highlight the practical value of using hprobiotics to enhance sow and piglet health in the face of significant disease challenges.

## Figures and Tables

**Figure 1 animals-15-00103-f001:**
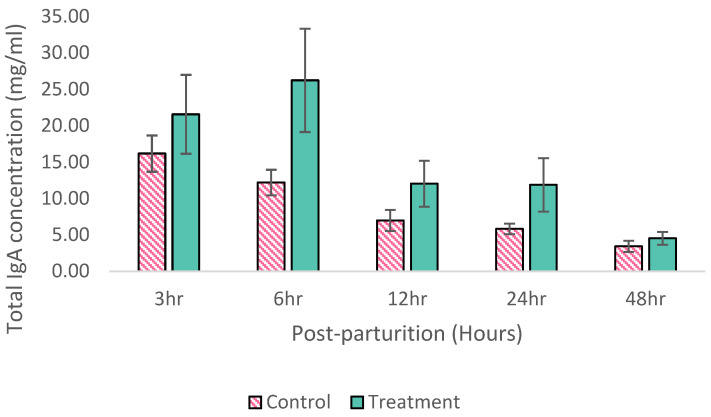
Immunoglobulin A (IgA) levels (means ± SE) in sow colostrum at 3, 6, 12, 24, and IgA in sow milk at 48 h after farrowing in control and the group of sows supplemented with probiotics; Treatment group.

**Figure 2 animals-15-00103-f002:**
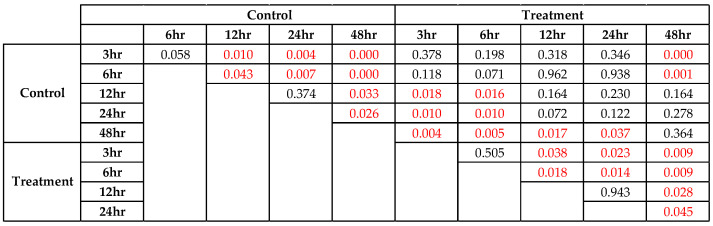
*p*-value of the total IgA levels in colostrum and milk across the different timepoints within and between the groups.

**Table 1 animals-15-00103-t001:** Ingredient composition of gestation and lactation diets given on an as-fed basis.

**Type of Diet**	**Gestation Diet**	**Lactation Diet**
Ingredient composition (%)		
Broken rice	31.50	28.35
Soybean meal (44% CP)	16.00	10.00
Full fat soybean meal (36% CP)	-	16.00
Rice bran	45.10	36.20
Fish meal (60% CP)	3.00	2.00
Rice bran oil	-	2.50
Dicalcium phosphate (18% P)	2.00	2.50
Limestone	1.60	1.50
DL-Methionine	0.08	0.05
L-Lysine	0.12	0.15
Salt	0.35	0.50
Premix	0.25	0.25
**Total**	**100.00**	**100.00**
Nutrient composition (%Dry matter Basis)		
Crude Protein	16.69	18.01
Metabolizable Energy (kcal/kg)	2995	3235
Calcium	1.08	1.02
Phosphorus	0.42	0.44
Methionine + Cystine	0.68	0.69
Lysine	0.85	0.96

**Table 2 animals-15-00103-t002:** Backfat thickness in sows supplemented with probiotics (means ± SE).

Parameter	Control	Treatment	*p*-Value
Backfat thickness at 1 month before farrowing	11.98 ± 0.15	11.83 ± 0.21	0.478
Backfat thickness at 3 weeks after farrowing	11.13 ± 0.17 ^a^	11.70 ± 0.14 ^b^	0.047

Different superscript lower-case letters indicate significant difference within rows (*p* < 0.05).

**Table 3 animals-15-00103-t003:** Productive performance in sows supplemented with probiotics (means ± SE).

Parameter	Control	Treatment	*p*-Value
Parity	3.0 ± 0.5	2.4 ± 0.5	0.625
Number of total born/L	13.1 ± 1.2	12.7 ± 0.9	0.552
Number born alive/L	11.3 ± 0.9	11.1 ± 0.9	0.867
Litter birth weight (kg)	15.5 ± 1.1	16.1 ± 1.2	0.830
Pre-weaning mortality rate per litter (%)	53.6 ± 11.3 ^a^	24.9 ± 9.4 ^b^	0.026
Number of weaned piglets/L	5.1 ± 1.2 ^a^	8.1 ± 1.0 ^b^	0.015
Weaning weight (kg)	3.9 ± 0.9 ^a^	5.9 ± 0.3 ^b^	0.032
Weaning weight gain (kg)	2.9 ± 0.7	4.5 ± 0.3	0.057

Different superscript lower-case letters indicate significant difference within rows (*p* < 0.05).

## Data Availability

Data are contained within the article.

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
