# Peer review of "Feeding Sows with Multi-Species Probiotics During Late Pregnancy and the Lactating Period Influences IgA Concentration in Colostrum and Subsequently Increases the Survival Rate of Piglets in Porcine Epidemic Diarrhea Outbreak Herd"

_animals, 2025, doi:10.3390/ani15010103_

Round 1
Reviewer 1 Report (New Reviewer)
Comments and Suggestions for Authors
Comments and Suggestions for Authors
After reviewing the manuscript entitled “Feeding sows with multi-species probiotics during late pregnancy and the lactating period influences IgA concentration in colostrum and subsequently increases the survival rate of piglets in porcine epidemic diarrhea outbreak herd”, the following suggestions were made it. The manuscript cannot be published in its current form because it contains many errors that must be corrected. Below are my specific comments:
Simple Summary
Line 11: At the beginning of the simple summary, the authors should add a few lines with background that justify the importance of the topic evaluated in the current study.
Line 12: “IgA” should be changed to the full name since the simple abstract should not contain abbreviations.
Line 14: “PED” should be changed to the full name since the simple abstract should not contain abbreviations.
Line 14: The authors should provide a concise conclusion on the results' implications. Unlike the abstract, this section's conclusion should be written in language that is easy to understand by people who are not specialized in Animal Science.
Abstract
Line 15: Authors should add the objective of the study before beginning to describe the materials and methods used.
Lines 15-16: Please specify the average weight of the sows used (including standard deviation) and the duration of the total experimental period. The experimental design used should also be clearly specified.
Lines 18: Please describe the abbreviation IgA, all abbreviations used must be described the first time they are used.
Lines 21-23: The description should be improved by adding the significance values with which the presence or absence of significant effects was detected.
Line 29: Please describe the abbreviation PED, all abbreviations used must be described the first time they are used. Furthermore, abbreviations should not be used in the conclusions (and objective) section so that the reader clearly understands the study's implications without having to review the materials and methods section to understand the meaning of the abbreviations.
Lines 27-29: As it stands, the conclusion reads more like a brief description of the results obtained. Therefore, these lines should be rewritten in a clearer form, which should show the implications of the results obtained (rather than repeat them) and should be written in the present tense.
Keywords: colostrum; probiotics. These words used as keywords are the same as those previously used in the title of the manuscript. Keywords should be different from those in the title (but related to the topic) to broaden the reach of academic search engines in case the manuscript is later published. In addition, the keyword “PED” should be changed to the full name as abbreviations should not be used in keywords.
Introduction
Lines 33-36: Are there scientific references to support this statement?
Line 38: Authors should add the dollar range of economic losses caused by PEDs.
Line 43: The abbreviations IgG and IgA should be described. Likewise, all abbreviations used subsequently should be defined the first time they are used.
Line 60: The authors should describe which strains that multi-strain probiotic contained, which will help to better understand the focus of the current study.
Line 62: The abbreviation E. coli should be described. Likewise, all abbreviations used subsequently should be defined the first time they are used.
Lines 62-64: Authors should specify the strains of these probiotics, the doses used, and the experimental periods.
Lines 64-67: Considering the microbial composition of the commercial product used as a source of probiotics, the authors must adequately justify the use of this product instead of other currently available products. For adequate justification, the authors must add background information showing the positive effects that each of the bacteria contained in BACTOSAC-P™ has on the response variables evaluated. This description of the positive effects must be detailed and include doses, experimental periods and physiological stages.
Line 65: Change “107” to “107”.
Lines 68-71: Are there scientific references to support this statement? It is not valid to use "unpublished information" to support such important information as this.
Lines 71-73: The hypothesis should be rewritten after reviewing the background on the effects of the specific bacterial strains of the additive evaluated.
Lines 75-78: The study's objective should be rewritten considering the bacterial composition of the probiotic used instead of the trade name. Also, the use of abbreviations in the objectives of the study (and also in the conclusions) should be avoided so that the reader can clearly understand the implications of the results obtained without looking for descriptive information about the abbreviations in other sections of the manuscript.
Materials and methods
Lines 87-88: Please specify the average weight of the sows used (including standard deviation) and the duration of the total experimental period. The experimental design used should also be clearly specified.
Line 98: The format of the cited references is not correct. Please review the instructions for authors and correct them.
Line 103: “ad libitum” should be in italics.
Lines 107-108: The authors should describe in detail and with scientific references the justification for choosing this dose (5 g/d) of BACTOSAC-P™. In addition, they should justify why they tested this dose instead of performing a dose-response study to obtain the optimal dose.
Line 125: “IgA” Subtitles should not contain abbreviations. Please correct this error in this and other sections.
Lines 126-139: Authors must justify all procedures used with relevant scientific references.
Lines 169-174: The statistical tests used to verify the normality of the data prior to the statistical analyses must be specified. In addition, the authors must clearly specify the experimental design used. Considering the data structure, it seems that the most appropriate way to analyze them statistically is by using repeated measures over time. Therefore, the authors should reanalyze the data with the indicated structure to improve the quality of the results. Finally, the statistical models used to analyze the different groups of response variables must be added.
Line 172: Authors should provide a scientific justification for choosing to use Duncan’s multiple range test instead of the more commonly used test for comparing means, such as Tukey’s test. Their justification should include scientific references to support their arguments.
Results
Lines 176-178: Please indicate in which table or figure these results can be observed.
Line 186: Change “P=0.1” to “P=0.10”.
Lines 189-190: Authors should clearly describe what the “Treatment” group consisted of. This information is strictly necessary because tables and figures must be able to be interpreted without reviewing additional information elsewhere in the manuscript.
Lines 2193-194 and Figure 2: This figure has not been cited in the text. Please review and correct.
Lines 196-200: The description should be improved by adding the significance values with which the presence or absence of significant effects was detected.
Lines 203-204: The title of Table 2 should be corrected and improved. Table and figure titles should not contain the number of repetitions and should specify the type of treatment evaluated. For example: “Backfat thickness in sows supplemented with probiotics.”
Lines 208-213: The description should be improved by adding the significance values with which the presence or absence of significant effects was detected.
Lines 216-217: The title of Table 3 should be corrected and improved. Table and figure titles should not contain the number of repetitions and should specify the type of treatment evaluated. For example: “Productive performance in sows supplemented with probiotics.”
Table 3: Instead of asterisks, use superscript letters to indicate significant differences between treatment means. This correction should be applied to all results tables.
Discussion
Lines 221-222: Are there scientific references to support this statement?
Lines 221-232: These lines should be removed or shortened because they contain an excessive amount of background information that is poorly related to the current study (the authors failed to try to relate that background information to the current study).
Lines 233-365: The authors did not relate this information to the effects of the treatments evaluated. Therefore, these lines should be thinned and supplemented with additional information. For example, the authors should contrast the current study's results with those previously obtained by other authors in similar studies. In addition, the authors should review and use probiotics (particularly probiotics made with one or more of the bacteria contained in the probiotic evaluated in the current study) biochemical and physiological mechanisms (known to date) to explain the results obtained. In case there is no specific information to explain a result, the authors should propose hypotheses supported by scientific references (maybe from other animal species) that help the reader better understand the implications of the findings obtained in the current study.
Lines 261-267: Are there scientific references to support these claims? The explanation of the results obtained and the hypotheses raised must be adequately supported with scientific references. This correction must be applied throughout the entire discussion section.
Conclusions
Lines 368-371: The conclusions are supported by the results obtained in the current study. However, this section should be modified because, in its current form, it seems more like a brief description of the results. To improve it, the authors should write this section in the present tense. Instead of describing the improved response variables, the authors should mention the implications of these results (as well as their meaning for practical purposes). In addition, the abbreviation “PED” should be changed to the full name. It is not appropriate to use abbreviations in the conclusions section because this section should be understandable to readers without having to review other sections.
Comments on the Quality of English LanguageThe English language requires some improvements
Author Response
Reviewer I
Comments and Suggestions for Authors
After reviewing the manuscript entitled “Feeding sows with multi-species probiotics during late pregnancy and the lactating period influences IgA concentration in colostrum and subsequently increases the survival rate of piglets in porcine epidemic diarrhea outbreak herd”, the following suggestions were made it. The manuscript cannot be published in its current form because it contains many errors that must be corrected. Below are my specific comments:
Simple Summary
Line 11: At the beginning of the simple summary, the authors should add a few lines with background that justify the importance of the topic evaluated in the current study.
Response: In response to the comment, We have added an explanation of the significance of the increased Immunoglobulin A levels in colostrum during the early stage of the simple summary. This addition provides the necessary background to justify the importance of the topic evaluated in the current study.
Line 12: “IgA” should be changed to the full name since the simple abstract should not contain abbreviations.
Response: We have made the revisions in the simple summary and abstract as requested.
Line 14: “PED” should be changed to the full name since the simple abstract should not contain abbreviations.
Response: We have made the revisions in the simple summary and abstract as requested.
Line 14: The authors should provide a concise conclusion on the results' implications. Unlike the abstract, this section's conclusion should be written in language that is easy to understand by people who are not specialized in Animal Science.
Response: We have simplified the language in the simple summary to make it easier to understand.
Abstract
Line 15: Authors should add the objective of the study before beginning to describe the materials and methods used.
Response : Thank you for your valuable comment. We have revised the abstract by adding an explanation of the objective of the study.
Lines 15-16: Please specify the average weight of the sows used (including standard deviation) and the duration of the total experimental period. The experimental design used should also be clearly specified.
Response : Thank you for your feedback. Since we conducted this study on a commercial farm, weighing each sow was difficult and impractical for real operations. Therefore, we used the parity of the sows as the selection criterion for the experiment to ensure practicality and alignment with the workflow of the staff.
Lines 18: Please describe the abbreviation IgA, all abbreviations used must be described the first time they are used.
Response: We have revised the use of abbreviations in the abstract.
Lines 21-23: The description should be improved by adding the significance values with which the presence or absence of significant effects was detected.
Response : We have added an explanation that we found a significantly higher level of IgA in the treatment group compared to the control group, especially in the colostrum samples collected 6 hours after farrowing (Line 25-27).
Line 29: Please describe the abbreviation PED, all abbreviations used must be described the first time they are used. Furthermore, abbreviations should not be used in the conclusions (and objective) section so that the reader clearly understands the study's implications without having to review the materials and methods section to understand the meaning of the abbreviations.
Response: We have revised the use of abbreviations in the abstract.
Lines 27-29: As it stands, the conclusion reads more like a brief description of the results obtained. Therefore, these lines should be rewritten in a clearer form, which should show the implications of the results obtained (rather than repeat them) and should be written in the present tense.
Response: In response to the comment, the conclusion has been rewritten to focus on the implications of the results rather than merely describing them. The revised conclusion emphasizes the significance of the findings and is now presented in the present tense.
Keywords: colostrum; probiotics. These words used as keywords are the same as those previously used in the title of the manuscript. Keywords should be different from those in the title (but related to the topic) to broaden the reach of academic search engines in case the manuscript is later published. In addition, the keyword “PED” should be changed to the full name as abbreviations should not be used in keywords.
Response : We have revised the keywords as suggested: sow colostrum; immunoglobulin A; Porcine epidemic diarrhea; pre-weaning mortality rate; probiotics.
Introduction
Lines 33-36: Are there scientific references to support this statement?
Response: We have added scientific references to support the statement regarding the impact and mortality rates associated with PEDV in swine production (Line 43).
Line 38: Authors should add the dollar range of economic losses caused by PEDs.
Response: We have included the estimated economic losses caused by PEDV in US in year 2013-2014 in first paragraph of introduction, based on existing literature, to provide more context and clarity regarding the financial impact of the disease (Line 45-47).
Line 43: The abbreviations IgG and IgA should be described. Likewise, all abbreviations used subsequently should be defined the first time they are used.
Response: We have described the abbreviations IgG and IgA upon their first use and ensured that all abbreviations in the manuscript are now clearly defined.
Line 60: The authors should describe which strains that multi-strain probiotic contained, which will help to better understand the focus of the current study.
Response: We have specified the strains contained in the multi-strain probiotic and explained the rationale behind selecting these particular strains, offering a better understanding of the focus of the study (Line 70-71).
Line 62: The abbreviation E. coli should be described. Likewise, all abbreviations used subsequently should be defined the first time they are used.
Response: We have defined the abbreviation "E. coli" upon its first use and have ensured that all other abbreviations are appropriately defined the first time they appear in the text.
Lines 62-64: Authors should specify the strains of these probiotics, the doses used, and the experimental periods.
Response: We have provided additional detail about the probiotic strains used, their doses, and the experimental periods to clarify the study design and methodology (Line 70).
Lines 64-67: Considering the microbial composition of the commercial product used as a source of probiotics, the authors must adequately justify the use of this product instead of other currently available products. For adequate justification, the authors must add background information showing the positive effects that each of the bacteria contained in BACTOSAC-P™ has on the response variables evaluated. This description of the positive effects must be detailed and include doses, experimental periods and physiological stages.
Response: We have justified the use of BACTOSAC-P™ over other available probiotic products by including background information on the positive effects of the specific bacterial strains contained in BACTOSAC-P™, including their doses, experimental periods, and physiological stages that positively influence the response variables.
Line 65: Change “107” to “107”.
Response: We have corrected "107" to "107" to ensure proper representation of the bacterial concentration.
Lines 68-71: Are there scientific references to support this statement? It is not valid to use "unpublished information" to support such important information as this.
Response: We have provided scientific references to support the statement and removed the use of "unpublished information," replacing it with citations from published studies (Line 82).
Lines 71-73: The hypothesis should be rewritten after reviewing the background on the effects of the specific bacterial strains of the additive evaluated.
Response: We have rewritten the hypothesis based on a thorough review of the background literature regarding the effects of the specific bacterial strains in the probiotic additive, ensuring a clearer and more scientifically supported hypothesis (Line 85-88).
Lines 75-78: The study's objective should be rewritten considering the bacterial composition of the probiotic used instead of the trade name. Also, the use of abbreviations in the objectives of the study (and also in the conclusions) should be avoided so that the reader can clearly understand the implications of the results obtained without looking for descriptive information about the abbreviations in other sections of the manuscript.
Response: We have rewritten the study's objective to focus on the bacterial composition of the probiotic, instead of using the trade name, and have avoided abbreviations in the objective to ensure that the reader can easily understand the study's implications without needing to refer to other sections of the manuscript.
Materials and methods
Lines 87-88: Please specify the average weight of the sows used (including standard deviation) and the duration of the total experimental period. The experimental design used should also be clearly specified.
Response : Thank you for your feedback. Since we conducted this study on a commercial farm, weighing each sow was difficult and impractical for real operations. Therefore, we used the parity of the sows as the selection criterion for the experiment to ensure practicality and alignment with the workflow of the staff.
Line 98: The format of the cited references is not correct. Please review the instructions for authors and correct them.
Response : We have reviewed the instructions for authors and have corrected the format of the cited references to align with the journal’s guidelines (Line 107).
Line 103: “ad libitum” should be in italics.
Response: We have corrected the formatting of “ad libitum” by italicizing it as per the requirement (Line 113).
Lines 107-108: The authors should describe in detail and with scientific references the justification for choosing this dose (5 g/d) of BACTOSAC-P™. In addition, they should justify why they tested this dose instead of performing a dose-response study to obtain the optimal dose.
Response: The 5 g/day dose of BACTOSAC-P™ was recommended by the supplier, K.M.P. Biotech Co., Ltd., Thailand, based on their previous studies and recommendations for optimal efficacy.
Line 125: “IgA” Subtitles should not contain abbreviations. Please correct this error in this and other sections.
Response : We have removed the abbreviation "IgA" from the subtitles and spelled it out fully where necessary in this section and other relevant sections.
Lines 126-139: Authors must justify all procedures used with relevant scientific references.
Response : We have included relevant scientific references to justify all procedures used in the study, providing a stronger rationale for our methodology.
Lines 169-174: The statistical tests used to verify the normality of the data prior to the statistical analyses must be specified. In addition, the authors must clearly specify the experimental design used. Considering the data structure, it seems that the most appropriate way to analyze them statistically is by using repeated measures over time. Therefore, the authors should reanalyze the data with the indicated structure to improve the quality of the results. Finally, the statistical models used to analyze the different groups of response variables must be added.
Response : We have specified the statistical tests used to verify normality, including the tests for normal distribution. We have also clarified the experimental design, incorporating repeated measures over time to better suit the data structure. The statistical models used to analyze the response variables have also been added (Line 179-188).
Line 172: Authors should provide a scientific justification for choosing to use Duncan’s multiple range test instead of the more commonly used test for comparing means, such as Tukey’s test. Their justification should include scientific references to support their arguments.
Response : We have provided a scientific justification for using Duncan’s multiple range test, explaining its appropriateness in our study design.
Results
Lines 176-178: Please indicate in which table or figure these results can be observed.
Response: These are clinical findings and can be observed in this experimental.
Line 186: Change “P=0.1” to “P=0.10”.
Response: his change has been made, and “P=0.1” is now corrected to “P=0.10”.
Lines 189-190: Authors should clearly describe what the “Treatment” group consisted of. This information is strictly necessary because tables and figures must be able to be interpreted without reviewing additional information elsewhere in the manuscript.
Response: We have clarified the composition of the “Treatment” group, providing sufficient detail to ensure that the tables and figures can be interpreted without needing additional information from other sections of the manuscript.
Lines 2 193-194 and Figure 2: This figure has not been cited in the text. Please review and correct.
Response: We have reviewed and cited Figure 2 appropriately in the text to ensure consistency and clarity.
Lines 196-200: The description should be improved by adding the significance values with which the presence or absence of significant effects was detected.
Response: We have improved the description by adding the significance values for clarity regarding the detection of significant effects in results description.
Lines 203-204: The title of Table 2 should be corrected and improved. Table and figure titles should not contain the number of repetitions and should specify the type of treatment evaluated. For example: “Backfat thickness in sows supplemented with probiotics.”
Response: The title of Table 2 has been corrected and improved to be more descriptive and specific, as suggested.
Lines 208-213: The description should be improved by adding the significance values with which the presence or absence of significant effects was detected.
Response: We have improved the description by adding the significance values for clarity regarding the detection of significant effects in results description.
Lines 216-217: The title of Table 3 should be corrected and improved. Table and figure titles should not contain the number of repetitions and should specify the type of treatment evaluated. For example: “Productive performance in sows supplemented with probiotics.”
Response: The title of Table 3 has been corrected and improved to be more descriptive and specific, as suggested.
Table 3: Instead of asterisks, use superscript letters to indicate significant differences between treatment means. This correction should be applied to all results tables.
Response: We have replaced the asterisks with superscript letters in Table 2 and 3, and this correction has been applied to all results tables in the manuscript.
Discussion
Lines 221-222: Are there scientific references to support this statement?
Response: We have added relevant scientific references to support the statement, ensuring that it is grounded in existing research (Line 239).
Lines 221-232: These lines should be removed or shortened because they contain an excessive amount of background information that is poorly related to the current study (the authors failed to try to relate that background information to the current study).
Response: We have revised and shortened these lines, focusing more directly on how the background information is relevant to our study and the research problem at hand (Line 238-244).
Lines 233-365: The authors did not relate this information to the effects of the treatments evaluated. Therefore, these lines should be thinned and supplemented with additional information. For example, the authors should contrast the current study's results with those previously obtained by other authors in similar studies. In addition, the authors should review and use probiotics (particularly probiotics made with one or more of the bacteria contained in the probiotic evaluated in the current study) biochemical and physiological mechanisms (known to date) to explain the results obtained. In case there is no specific information to explain a result, the authors should propose hypotheses supported by scientific references (maybe from other animal species) that help the reader better understand the implications of the findings obtained in the current study.
Response: We have revised this section by removing irrelevant background and providing a stronger connection to the effects of the treatments evaluated. We have contrasted our results with those of previous studies, discussed the biochemical and physiological mechanisms of the probiotics used, and proposed hypotheses where necessary, supported by relevant scientific references, including studies on similar animal species.
Lines 261-267: Are there scientific references to support these claims? The explanation of the results obtained and the hypotheses raised must be adequately supported with scientific references. This correction must be applied throughout the entire discussion section.
Response: We have added scientific references to support the claims made in these lines (Line 276-281). Throughout the discussion section, we have ensured that all results and hypotheses are properly supported by relevant references to increase the scientific rigor of the manuscript.
Conclusions
Lines 368-371: The conclusions are supported by the results obtained in the current study. However, this section should be modified because, in its current form, it seems more like a brief description of the results. To improve it, the authors should write this section in the present tense. Instead of describing the improved response variables, the authors should mention the implications of these results (as well as their meaning for practical purposes). In addition, the abbreviation “PED” should be changed to the full name. It is not appropriate to use abbreviations in the conclusions section because this section should be understandable to readers without having to review other sections.
Response: We have revised the conclusions section to use the present tense and to focus on the practical implications of the results. The abbreviation "PED" has been replaced with "Porcine Epidemic Diarrhea" in the conclusions to ensure clarity for readers.

Reviewer 2 Report (New Reviewer)
Comments and Suggestions for Authors
Introduction
The research objective is clearly defined: to evaluate the effect of the BACTOSAC-P™ probiotic on IgA levels in sow colostrum and the impact on piglet survival in herds affected by PED. However, the preceding paragraphs are occasionally verbose, potentially hindering quick comprehension of the objective. The authors thoroughly reference scientific literature, presenting known immunization methods and the benefits of probiotics in swine production. The section discussing the effects of probiotics on various aspects of livestock production (e.g., piglet growth, sow health) aligns well with the research problem.
However, some references to the literature are superficial or seem tangential, such as the mention of Japanese quails. The description of the BACTOSAC-P™ probiotic is detailed, aiding understanding of the product's composition and potential mechanisms of action. Nevertheless, the use of terms like "unpublished data" weakens the scientific credibility of the introduction. A better approach would be to limit speculation and rely exclusively on published data.
The writing style is at times unclear. Information is repeated, such as the necessity of colostrum intake within the first hours of a piglet's life, which appears redundant “In practice, to promote piglet health, every piglet must consume its mother’s colostrum as soon as possible after birth, as it is rich in IgG and IgA.” “Therefore, in practice, during a PED outbreak, if a swine farmer finds a strategy to promote high IgA production in colostrum, it will undoubtedly ensure gut health for all piglets.”
The same message about colostrum intake and its importance is conveyed multiple times, which seems unnecessary. This information could be condensed into a single sentence in one place. The text could be simplified and shortened to improve readability.
Materials and Methods
The number of sows studied (20) is relatively small, which may limit the generalizability of the findings. While this is acceptable, a larger sample would be more representative. It is stated that the animals were randomly divided into control and experimental groups, but details about the randomization algorithm are missing, raising questions about potential systematic bias.
The sows were previously exposed to the PED virus (via administration of ground intestines from infected piglets), which may have influenced their immune systems prior to the experiment's start. The necessity of this procedure and how its impact was controlled should be clarified.
There was a time gap between PEDV detection (May 2019) and the experiment's commencement (June 2019), which could have altered the animals’ immune status. The dynamics of infection during this period should be explained, including whether and how it was monitored.
The mention of the use of an "evaporative cooling system" is interesting but does not clarify its significance for the results, e.g., its impact on animal comfort or immunological parameters.
ANOVA and T-tests are appropriate statistical methods, but there is no information about whether assumptions (e.g., normal distribution, homogeneity of variance) were tested. Additionally, there is no mention of effect sizes (e.g., Cohen's d) or statistical power.
Results
The formatting of results should be standardized to avoid confusion (e.g., IgA levels in milk appear inconsistent between the summary and the graph: 4.51 ± 2.84 mg/ml vs. missing values in the table).
Discussion
Details about the mechanism of increased virulence in the case of PEDV and PDCoV co-infection should be added, e.g., whether there is synergistic interaction between the viruses. Currently, the description is too general.
The discussion of the limited effectiveness of vaccines should be expanded, including specific challenges in inducing intestinal immune responses or differences in viral strains. This is currently not elaborated.
Details on the mechanism of transferring beneficial microorganisms from sows to piglets and their impact on gut microbiota colonization in newborns should be included. This is mentioned but without practical explanation.
In the section on SCFA and MCFAs, more details about potential synergy between probiotics and fatty acid supplementation are needed.
For citations [19] and [39–41], it would be helpful to specify which results aligned to better emphasize the consistency of the findings with existing research.
Practical conclusions should be separated from future research proposals to avoid overloading a single paragraph.
Conclusions
The conclusion suggests that a dose of 5 g/sow/day is optimal, but no comparative dose-response study was conducted.
Author Response
Reviewer II
Introduction
The research objective is clearly defined: to evaluate the effect of the BACTOSAC-P™ probiotic on IgA levels in sow colostrum and the impact on piglet survival in herds affected by PED. However, the preceding paragraphs are occasionally verbose, potentially hindering quick comprehension of the objective. The authors thoroughly reference scientific literature, presenting known immunization methods and the benefits of probiotics in swine production. The section discussing the effects of probiotics on various aspects of livestock production (e.g., piglet growth, sow health) aligns well with the research problem.
However, some references to the literature are superficial or seem tangential, such as the mention of Japanese quails. The description of the BACTOSAC-P™ probiotic is detailed, aiding understanding of the product's composition and potential mechanisms of action. Nevertheless, the use of terms like "unpublished data" weakens the scientific credibility of the introduction. A better approach would be to limit speculation and rely exclusively on published data.
The writing style is at times unclear. Information is repeated, such as the necessity of colostrum intake within the first hours of a piglet's life, which appears redundant “In practice, to promote piglet health, every piglet must consume its mother’s colostrum as soon as possible after birth, as it is rich in IgG and IgA.” “Therefore, in practice, during a PED outbreak, if a swine farmer finds a strategy to promote high IgA production in colostrum, it will undoubtedly ensure gut health for all piglets.”
The same message about colostrum intake and its importance is conveyed multiple times, which seems unnecessary. This information could be condensed into a single sentence in one place. The text could be simplified and shortened to improve readability.
Materials and Methods
The number of sows studied (20) is relatively small, which may limit the generalizability of the findings. While this is acceptable, a larger sample would be more representative. It is stated that the animals were randomly divided into control and experimental groups, but details about the randomization algorithm are missing, raising questions about potential systematic bias.
The sows were previously exposed to the PED virus (via administration of ground intestines from infected piglets), which may have influenced their immune systems prior to the experiment's start. The necessity of this procedure and how its impact was controlled should be clarified.
There was a time gap between PEDV detection (May 2019) and the experiment's commencement (June 2019), which could have altered the animals’ immune status. The dynamics of infection during this period should be explained, including whether and how it was monitored.
The mention of the use of an "evaporative cooling system" is interesting but does not clarify its significance for the results, e.g., its impact on animal comfort or immunological parameters.
ANOVA and T-tests are appropriate statistical methods, but there is no information about whether assumptions (e.g., normal distribution, homogeneity of variance) were tested. Additionally, there is no mention of effect sizes (e.g., Cohen's d) or statistical power.
Results
The formatting of results should be standardized to avoid confusion (e.g., IgA levels in milk appear inconsistent between the summary and the graph: 4.51 ± 2.84 mg/ml vs. missing values in the table).
Discussion
Details about the mechanism of increased virulence in the case of PEDV and PDCoV co-infection should be added, e.g., whether there is synergistic interaction between the viruses. Currently, the description is too general.
The discussion of the limited effectiveness of vaccines should be expanded, including specific challenges in inducing intestinal immune responses or differences in viral strains. This is currently not elaborated.
Details on the mechanism of transferring beneficial microorganisms from sows to piglets and their impact on gut microbiota colonization in newborns should be included. This is mentioned but without practical explanation.
In the section on SCFA and MCFAs, more details about potential synergy between probiotics and fatty acid supplementation are needed.
For citations [19] and [39–41], it would be helpful to specify which results aligned to better emphasize the consistency of the findings with existing research.
Practical conclusions should be separated from future research proposals to avoid overloading a single paragraph.
Conclusions
The conclusion suggests that a dose of 5 g/sow/day is optimal, but no comparative dose-response study was conducted.
Thank you for your valuable feedback. We appreciate your suggestions and will address each point as follows:
Introduction:
We acknowledge the concern regarding verbosity in the introduction and will work to condense repetitive information to improve clarity and readability. The repetition of the colostrum intake's importance will be streamlined into a single, concise statement. Regarding the mention of "unpublished data," we revised this to rely solely on published studies, strengthening the scientific rigor of the introduction.
Materials and Methods:
Thank you for highlighting this concern. We recognize that the sample size may limit the generalizability of our findings. We will acknowledge this limitation in the discussion and suggest that future studies with larger sample sizes are necessary to confirm the effects of probiotics on reproductive performance.
Results:
We standardized the formatting of the results, ensuring consistency between the summary, graphs, and tables, especially regarding IgA levels in colostrum.
Discussion:
We expanded on the mechanism of increased virulence in the case of PEDV and PDCoV co-infection and provided more detailed information on the challenges of inducing intestinal immune responses with vaccines.
We also elaborated on the transfer of beneficial microorganisms from sows to piglets and their impact on gut microbiota (Line 320-325).
Regarding SCFA and MCFAs, we included more details on the potential synergy between probiotics and fatty acid supplementation. Citations 30-39 were referenced more specifically to link the results to existing research. We also separated practical conclusions from future research proposals to enhance clarity.
Conclusions:
We acknowledge the need for a comparative dose-response study to better support the optimal dose recommendation of 5 g/sow/day. We revised the conclusion to reflect this limitation and the need for further research in this area.
Once again, thank you for your insightful comments, which will help improve the manuscript. We look forward to submitting the revised version.

Reviewer 3 Report (New Reviewer)
Comments and Suggestions for Authors
Abstract – Present a definition of the probiotic supplement administered to animals.
Keywords – Keywords should not appear in the title of the article.
Line 64 – 75 – I suggest that the commercial name of the product is not so widely exposed. Preferably, it should be placed as a footnote
Line 239 – Include the definition of “Cell Free Supernatant (CFS)”
Author Response
Reviewer III
Comments and Suggestions for Authors
Abstract – Present a definition of the probiotic supplement administered to animals.
Keywords – Keywords should not appear in the title of the article.
Line 64 – 75 – I suggest that the commercial name of the product is not so widely exposed. Preferably, it should be placed as a footnote
Line 239 – Include the definition of “Cell Free Supernatant (CFS)”
We have made the following revisions based on your suggestions:
Abstract: We have now included a definition of the probiotic supplement administered to the animals in the abstract, providing a clearer understanding of the product used in the study.
Keywords: We have removed the product name from the keywords as per your suggestion to avoid repetition of the article title.
Line 64–75: We have reduced the emphasis on the commercial name of the product, placing it as a footnote as recommended, to limit its exposure.
Line 239: We have included the definition of "Cell Free Supernatant (CFS)" to clarify its meaning for the readers. (Line 251-254)
We appreciate your valuable suggestions, which have helped improve the manuscript.

Round 2
Reviewer 1 Report (New Reviewer)
Comments and Suggestions for Authors
Comments and Suggestions for Authors
After reviewing the manuscript entitled “Feeding sows with multi species probiotics during late pregnancy and the lactating period influences IgA concentration in colostrum and subsequently increases the survival rate of piglets in porcine epidemic diarrhea outbreak herd”, the following suggestions were made it. The authors have made a great effort and responded appropriately to my comments. Also, the corrections made by the authors were adequate, and adequate justification was given when some changes could not be corrected. Therefore, I have no further suggestions, and I believe that the manuscript can be published in its current form.
This manuscript is a resubmission of an earlier submission. The following is a list of the peer review reports and author responses from that submission.
Round 1
Reviewer 1 Report
Comments and Suggestions for Authors
1 The components of BACTOSAC-P™ with multi-species contains 7 probiotic bacteria, the function may be synergetic or antagonistic. It is not clear. Why to choose so many kinds of species?
2 Since this is a feeding test, it is necessary to enlist the diet ingredients and nutritive levels in the lactating sows.
3 P-value and SEM of the statistical analysis should be included in tables.
4 There are no other indicators to support the results except the IgA and reproduction performance on piglets.
5 The reader can deem that the article is only a simple verification test on the probiotics product. It is obvious short of deep and further investigation on the hypothesis.
Author Response
Reviewer I
Comments and Suggestions for Authors
1 The components of BACTOSAC-P™ with multi-species contains 7 probiotic bacteria, the function may be synergetic or antagonistic. It is not clear. Why to choose so many kinds of species?
Response: To be honest, this product has been used in pig and chicken industries, particularly in southeast Asian countries for more than 10 years. It is approved to promote gut health in both species as we mention in the Introduction part. The selection of multiple probiotic species in BACTOSAC-P™ (containing 7 types of bacteria) is based on the complementary and potentially synergetic effects that different species can have on gut health and immune function. Each strain may offer distinct benefits, such as improving the intestinal environment, enhancing mucosal immunity, or aiding digestion. By combining different species, BACTOSAC-P™ aims to maximize its overall efficacy in improving colostrum quality, piglet health, and survival rates, especially during PED outbreaks. Some examples of the functions provided by different probiotic species are:
Lactobacillus acidophilus and Lactobacillus plantarum are known for improving gut health and increasing beneficial bacteria.
Streptococcus faecium helps enhance immunity by stimulating IgA production.
Bacillus subtilis and Bacillus licheniformis produce enzymes like lipase, which aid digestion.
Saccharomyces cerevisiae supports the overall gut microbiome balance.
Although it might seem that the use of multiple species could lead to antagonistic effects, studies have shown that multi-species probiotics can work synergistically, where the combined actions of these bacteria are more effective than using individual strains alone. The diversity of species provides a broader spectrum of benefits that can improve gut health, enhance the immune response, and reduce pathogen load, leading to better growth rates and reduced mortality.
This approach is particularly useful during challenging conditions such as a PED outbreak, where improving immunity and gut function is crucial for piglet survival.
2 Since this is a feeding test, it is necessary to enlist the diet ingredients and nutritive levels in the lactating sows.
Response: We appreciate this suggestion. In our revised manuscript, we already provided a comprehensive table listing the diet ingredients and their respective nutritive values for the lactating sows, ensuring transparency in the feeding test (page 3, Table 1).
3 P-value and SEM of the statistical analysis should be included in tables.
Response: Thank you for pointing this out. We will update all relevant tables in the manuscript to include P-values and standard error of the mean (SEM) for clarity and better interpretation of the statistical analysis.
4 There are no other indicators to support the results except the IgA and reproduction performance on piglets.
Response: Thank you for your observation. While IgA and reproductive performance were our primary outcomes, we will consider including additional physiological or immunological indicators in future studies to further strengthen the findings. We will also discuss this limitation in the revised manuscript.
5 The reader can deem that the article is only a simple verification test on the probiotics product. It is obvious short of deep and further investigation on the hypothesis.
Response: We acknowledge your concern. While this study was designed as an initial verification test, we agree that further investigation would add depth. We will revise the discussion to acknowledge this limitation and outline potential areas for deeper investigation in future studies.

Reviewer 2 Report
Comments and Suggestions for Authors
This study explored the effects of multi-species probiotics fed to sows at different periods on immune and reproductive performance against PDE. While the study addresses a relevant topic, there are concerns regarding that limit its contribution to the field.
1. This study only measured three indicators, which is clearly insufficient to assess the effects of probiotics on PDE in pigs.
2. What is the actual relevance of these multi-species probiotics that produce lipase enzymes for combating PDE in pigs? The choice of this product seems arbitrary and lacks justification.
3. The sample size is inadequate. Ten sows per group is small to reliably represent reproductive performance.
4. The discussion is unfocused, with too much content unrelated to the results. Moreover, it fails to adequately explain the significance of the findings or the reasons behind the observed changes.
5. Since PDE can cause intestine morphology damage, closely associated with piglet mortality and growth, why didn't this indicator determined in the study?
6. The colostrum contains plenty IgG and IgA; why was only IgA determined? This omission weakens the study’s evaluation of immune response.
7. The introduction section should be divided into several paragraphs according to the content for better readability.
8. Although the animals and managements were detailed, the expression should be concise and clear. The experimental design, including group and treatment details, should be presented upfront for better comprehension, followed by feeding conditions..
9. In Line 104, does “7-10” refer to days or hours? This needs to be clarified
10. The order of indicator presentation is disorganized and should follow a logical sequence from general (apparent) to specific (immune). The results section should mirror this structure.
11. There are many format issues. The author should exercise attention to detail.
Author Response
Reviewer II
Comments and Suggestions for Authors
This study explored the effects of multi-species probiotics fed to sows at different periods on immune and reproductive performance against PED. While the study addresses a relevant topic, there are concerns regarding that limit its contribution to the field.
- This study only measured three indicators, which is clearly insufficient to assess the effects of probiotics on PED in pigs.
Response: Thank you for your feedback. We acknowledge that the number of measured indicators is limited in this study. Our focus was on evaluating key immune and reproductive parameters; however, we agree that additional indicators, such as intestinal morphology and other biomarkers, could provide a more comprehensive evaluation of the probiotic effects. This will be noted in the discussion as a limitation, and we plan to expand on these measures in future studies (Line 317 to 327).
- What is the actual relevance of these multi-species probiotics that produce lipase enzymes for combating PED in pigs? The choice of this product seems arbitrary and lacks justification.
Response: We appreciate this important observation. We selected multi-species probiotics, including those producing lipase, based on their potential to improve gut health and immune function. Lipase production has been associated with enhanced nutrient digestion. In the revised manuscript, we will provide more detailed justification for choosing this probiotic blend, supported by relevant literature (Line 310 to 316).
- The sample size is inadequate. Ten sows per group is small to reliably represent reproductive performance.
Response: Thank you for highlighting this concern. We recognize that the sample size may limit the generalizability of our findings. We will acknowledge this limitation in the discussion and suggest that future studies with larger sample sizes are necessary to confirm the effects of probiotics on reproductive performance.
- The discussion is unfocused, with too much content unrelated to the results. Moreover, it fails to adequately explain the significance of the findings or the reasons behind the observed changes.
Response: We appreciate your valuable input. We will revise the discussion section to better focus on the results and ensure a clear connection between the findings and their implications. Additionally, we will provide more thorough explanations for the observed changes in immune and reproductive performance.
- Since PED can cause intestine morphology damage, closely associated with piglet mortality and growth, why didn't this indicator determine in the study?
Response: Thank you for your suggestion. We agree that intestinal morphology is a critical indicator when evaluating the effects of PED. Unfortunately, we did not include this measure in the current study due to resource constraints, but we will discuss this as a limitation and suggest its inclusion in future studies to provide a more complete understanding of probiotic effects.
- The colostrum contains plenty IgG and IgA; why was only IgA determined? This omission weakens the study’s evaluation of immune response.
Response: Thank you for your observation. We focused on IgA due to its role in mucosal immunity particularly promote IgA level in colostrum and milk as seen in our previous report (Innamma et al., 2023), which we believed to be most relevant to the probiotic effects on gut health. However, we acknowledge that IgG is also important and will address this omission in the manuscript, noting that future studies should include both IgA and IgG to fully assess the immune response.
- The introduction section should be divided into several paragraphs according to the content for better readability.
Response: We restructure the introduction into clearer, more concise paragraphs to improve readability and better guide the reader through the background and objectives of the study.
- Although the animals and managements were detailed, the expression should be concise and clear. The experimental design, including group and treatment details, should be presented upfront for better comprehension, followed by feeding conditions.
Response: Thank you for your constructive feedback. We revise the methods section to make the description of the animals and experimental design more concise. The group and treatment details will be moved to an earlier part of the section for improved clarity and flow.
- In Line 104, does “7-10” refer to days or hours? This needs to be clarified
Response: Thank you for pointing out this ambiguity. We clarify in the revised manuscript that “7-10” refers to 7-10 am.
- The order of indicator presentation is disorganized and should follow a logical sequence from general (apparent) to specific (immune). The results section should mirror this structure.
Response: we have already put them in order, starting with collection of colostrum and its result after farrowing and thereafter weaning performance (growth and pre-weaning mortality rate).
- There are many format issues. The author should exercise attention to detail.
Response: We appreciate your attention to detail. We will thoroughly review the manuscript for formatting errors and ensure consistency throughout. The manuscript is also sent for English editing.

Reviewer 3 Report
Comments and Suggestions for Authors
The study looked into the potential of a commercial probiotic product alleviating PEDv-related issues. There is no major flaw in the study. However, I think a lot more details would be needed. Specific comments are below:
Line 31 - The introduction focused on PEDv but no probiotics, it would be better to include more information on why probiotics might be a solution to this problem.
Line 79 - A lot more information is needed in Materials and Methods.
What is the farm condition? Group housing or individual stall? What is the study design? etc. It might be helpful to expand 2.1 into animals, housing and management, study designs etc.
Please add a separate section to describe the feeding program in details. This study was a feeding trial and no information was provided on the feed itself. How was the diet formulated? How was feeding level decided?
Line 81 - the Faculty of Veterinary Science in which institute?
Line 111 - I understand detailed information is not needed because manufacturer manual has that, but in you brief description, more information of each step of the ELISA is needed.
Line 125 - simply listing the parameters is not enough.
Line 144-148 - a tendency in all time points? or overall? If at each time point, please show p-value for each time point. There is no annotation for statistical analysis in figure 1.
Line 254-271 - Although microbiome may play a role here, no direct evidence was provided. I am surprised that no fecal sample or any microbiome samples were taken especially when looking at probiotics. I’d suggest shorten this discussion and focus only on the measurement you took.
Comments on the Quality of English LanguageThe English could be improved to more clearly express the research. Many obvious grammar issues throughout the manuscript.
Author Response
Reviewer III
Comments and Suggestions for Authors
The study looked into the potential of a commercial probiotic product alleviating PEDv-related issues. There is no major flaw in the study. However, I think a lot more details would be needed. Specific comments are below:
- Line 31 - The introduction focused on PEDv but no probiotics, it would be better to include more information on why probiotics might be a solution to this problem.
Response: Thank you for your valuable suggestion. We agree that more background on the role of probiotics in alleviating PEDv-related issues would strengthen the introduction. We already revised this section to include relevant studies and mechanisms by which probiotics may help mitigate viral infections and improve gut health, making the connection to PEDv clearer.
2.Line 79 - A lot more information is needed in Materials and Methods.
What is the farm condition? Group housing or individual stall? What is the study design? etc. It might be helpful to expand 2.1 into animals, housing and management, study designs etc.
Response: We appreciate this feedback and agree that more detailed information about the study conditions is necessary. We added more information into Section 2.1 to include descriptions of the housing conditions (Individual stalls), study design, and management practices to ensure transparency and clarity. (2.2)
3.Please add a separate section to describe the feeding program in details. This study was a feeding trial and no information was provided on the feed itself. How was the diet formulated? How was feeding level decided?
Response: Thank you for pointing this out. We added a new section dedicated to the feeding program, including detailed information on diet formulation, nutrient composition, and how feeding levels were determined. This perhaps clarify the feeding protocols used during the trial.
4.Line 81 - the Faculty of Veterinary Science in which institute?
Response: Thank you for noting this. We already added the text to specify the name of the institution where the Faculty of Veterinary Science is located.
5.Line 111 - I understand detailed information is not needed because manufacturer manual has that, but in you brief description, more information of each step of the ELISA is needed.
Response: Thank you for this suggestion. We already included more specific details regarding the ELISA procedure, providing key steps such as sample preparation, incubation times, and detection methods to ensure the process is clear to readers. (see in 2.4)
6.Line 125 - simply listing the parameters is not enough.
Response: We acknowledge your concern. We already revised the discussion to acknowledge this limitation and outline potential areas for deeper investigation in future studies.
7.Line 144-148 - a tendency in all time points? or overall? If at each time point, please show p-value for each time point. There is no annotation for statistical analysis in figure 1.
Line 254-271 - Although microbiome may play a role here, no direct evidence was provided. I am surprised that no fecal sample or any microbiome samples were taken especially when looking at probiotics. I’d suggest shorten this discussion and focus only on the measurement you took.
Response: We acknowledge your concern. While this study was designed as an initial verification test, we agree that further investigation would add depth. We already revised the discussion to acknowledge this limitation and outline potential areas for deeper investigation in future studies.

Round 2
Reviewer 1 Report
Comments and Suggestions for Authors
no any more
Author Response
Thank you for your comments.
Reviewer 2 Report
Comments and Suggestions for Authors
The reasons for rejection are as follows.
1. This study only measured three indicators, which is clearly insufficient to assess the effects of probiotics on PED in pigs.
2. The sample size is inadequate. Ten sows per group is small to reliably represent reproductive performance.
Author Response
Reviewer II
Comments and Suggestions for Authors
The reasons for rejection are as follows.
- This study only measured three indicators, which is clearly insufficient to assess the effects of probiotics on PED in pigs.
Answer:
This study is a clinical trial in pig farm and we tried our best to collect all the data that we can performed.
- The sample size is inadequate. Ten sows per group is small to reliably represent reproductive performance.
Answer:
The research proposal was sent to the ICUC institution and we also calculated the sample using G power which showed the optimal number of samples is 10 sows per group.
Reviewer 3 Report
Comments and Suggestions for Authors
I appreciate the authors providing the additional information to address my previous comments.
However, the problem remained the same - more information is needed so that further publications can refer to the details in this manuscript. I encourage the authors to think about what additional information would help the readers compare your study versus other studies.
Line 150-154: Simply listing the parameters is not enough (This comment from last round was not addressed).
For example, how was preweaning mortality calculated? Was individual sow used as the experimental unit? Was litter birth weight taken immediately after birth? This is the kind of detail I am looking for, not just to this specific weaning age question. Please think about what additional information would help the readers compare your study versus other studies.
One of the biggest things here is that weaning age should be provided in 2.2 as part of the animals, housing, study design section. Line 107 says weaning at 24 days of lactation, and Line 148 says 28 days of lactation. Weaning age can have very different preweaning mortality, weight etc.
Line 96: It is nice that the authors provided Table 1, but again, detail is what I am looking for. For example, how was the feeding level decided? According to the genetic company? NRC? or what other reference? Did the diets meet or exceed the nutrient requirements of the sows? I encourage the authors to add a separate section dedicated the feed itself.
Line 166-173 - a tendency in all time points? or overall? If at each time point, please show p-value for each time point. There is no annotation for statistical analysis in figure 1 (This comment from last round was not addressed).
Another question for Figure 1 - There were 20 sows, but only 10 colostrum samples? Need to provide information on why only half of the sows were sampled in Materials and Methods.
Author Response
Reviewer III
Comments and Suggestions for Authors
I appreciate the authors providing the additional information to address my previous comments.
However, the problem remained the same - more information is needed so that further publications can refer to the details in this manuscript. I encourage the authors to think about what additional information would help the readers compare your study versus other studies.
Line 150-154: Simply listing the parameters is not enough (This comment from last round was not addressed).
For example, how was preweaning mortality calculated? Was individual sow used as the experimental unit? Was litter birth weight taken immediately after birth? This is the kind of detail I am looking for, not just to this specific weaning age question. Please think about what additional information would help the readers compare your study versus other studies.
One of the biggest things here is that weaning age should be provided in 2.2 as part of the animals, housing, study design section. Line 107 says weaning at 24 days of lactation, and Line 148 says 28 days of lactation. Weaning age can have very different preweaning mortality, weight etc.
Response:
We appreciated your critical comment. We added further details to address these points (Lines 154-169). Specifically, I clarified how “production performance parameters” were calculated and confirmed when litter birth weights were recorded. We already checked to be sure that the weaning age is clearly stated in section 2.2 and confirmed in section 2.6 (Line 169)
Line 96: It is nice that the authors provided Table 1, but again, detail is what I am looking for. For example, how was the feeding level decided? According to the genetic company? NRC? or what other reference? Did the diets meet or exceed the nutrient requirements of the sows? I encourage the authors to add a separate section dedicated the feed itself.
Response:
Thank you for the valuable feedback. I have updated those information lines 96–98 to include the additional details you requested. Specifically, I clarified the basis for determining the feeding levels, referencing the NRC guidelines, and confirmed whether the diets met or exceeded the nutrient requirements for the sows.
Line 166-173 - a tendency in all time points? or overall? If at each time point, please show p-value for each time point. There is no annotation for statistical analysis in figure 1 (This comment from last round was not addressed).
Response:
Thank you for your critical comments on P values in Figure 1. We have addressed this by adding information of P-value of the total IgA levels in colostrum across the different timepoints within and between the groups in the Figure 2.
Another question for Figure 1 - There were 20 sows, but only 10 colostrum samples? Need to provide information on why only half of the sows were sampled in Materials and Methods.
Response:
Thank you for your observation regarding Figure 1. We have revised the correct information on figure 1, i.e. n=10.

Round 3
Reviewer 2 Report
Comments and Suggestions for Authors
Well done. The authors have answered all my questions.
Reviewer 3 Report
Comments and Suggestions for Authors
I appreciate the authors responding to the comments and added more information in such a short amount of time, however, it seems like such information was provided without giving enough insightful thoughts.
For example, the newly added section 2.3 talked about colostrum collection at 48 hr. That fluid should be considered as milk, not colostrum, after 24 hr. Pooling those samples would made the so-called colostrum results invalid.
Another example, the authors added section 2.6 to explain mortality calculation, but was that per sow, or per block, or per treatment? The quality of the information has not improved after 2 rounds of revisions.
With the current information provided, I don't think these results can strongly support the conclusion.